# Defect scattering can lead to enhanced phonon transport at nanoscale

Yue Hu[1,2], Jiaxuan Xu[1,3], Xiulin Ruan [4] & Hua Bao [1,3] ✉

Defect scattering is well known to suppress thermal transport. In this study, however, we perform both molecular dynamics and Boltzmann transport equation calculations, to demonstrate that introducing defect scattering in nanoscale heating zone could surprisingly enhance thermal conductance of the system by up to 75%. We further reveal that the heating zone without defects yields directional nonequilibrium with overpopulated oblique-propagating phonons which suppress thermal transport, while introducing defects redirect phonons randomly to restore directional equilibrium, thereby enhancing thermal conductance. We demonstrate that defect scattering can enable such thermal transport enhancement in a wide range of temperatures, materials, and sizes, and offer an unconventional strategy for enhancing thermal transport via the manipulation of phonon directional nonequilibrium.

Engineering thermal transport has long been an important scientific endeavor with fundamental and practical interests[1]. Defects in materials are well known to introduce excessive scattering and suppress thermal transport[2–4]. Defect engineering is thus widely applied for developing low thermal conductivity materials[5,6]. On the other hand, when seeking high thermal conductivity materials, reducing defect scattering is sought, usually by growing high-quality crystalline structures or through isotope purification[7,8].

In this study, we find that, counterintuitively, defect scattering could enhance thermal conductivity at nanoscale. We first adopt molecular dynamics (MD) simulations to demonstrate that introducing defect scattering in the heating zone can enhance thermal transport in the ballistic regime, i.e., when the characteristic length is smaller than the phonon mean free path. With an analysis based on the phonon Boltzmann transport equation (BTE), we show that the defect-free volumetric heating zone overpopulates oblique-propagating phonons, while introducing defects would redirect phonons randomly to restore directional equilibrium, thereby enhancing thermal conductance. Such a mechanism exists for a wide range of temperatures, sizes, and materials. Such findings provide new insights into directional nonequilibrium phonon transport and can extend our current strategies for manipulating thermal transport.

## Results and discussion
### Defect scattering leads to enhanced phonon transport
In this study, we consider a typical one-dimensional system where a thin film is sandwiched between an adiabatic boundary and a heat sink, as shown in Fig. 1a. Heat generation occurs in a thin heating zone near the adiabatic boundary. Note that such a system is quite similar to a transistor[9], in which heat is generated in a nanometer-thin channel region by electron scattering. To model such a system, we first adopt the MD simulation. The advantage of MD is that it includes almost all the nanoscale thermal transport physics, except for its classical nature[10,11]. Figure 1b illustrates the simulation cell of the system in MD simulations. The phonons gain energy from the heating zone and then carry the energy to the heat sink. The regions from the top to the bottom are the fixed atoms (preventing sublimating), heating zone, substrate, fixed temperature heat sink, and fixed atoms, respectively. For demonstration purposes, the substrate is pure Si and the heating zone is Si with Ge impurities occupying random sites. The interatomic potential is the Tersoff potential[12]. After a convergence test (see Supplementary Note S1), a cross-sectional area of $8 \times 8$ -unit cells is considered and periodic boundary conditions are applied in lateral directions. In the previous study, we have demonstrated that the Nose-Hoover chain thermostat behaves like a spatially uniform heat generation and the Langevin thermostat behaves like a fixed temperature

[1]University of Michigan–Shanghai Jiao Tong University Joint Institute, Shanghai Jiao Tong University, Shanghai 200240, PR China. [2]CTG Wuhan Science and Technology Innovation Park, China Three Gorges Corporation, Wuhan 430010, PR China. [3]Global Institute of Future Technology, Shanghai Jiao Tong University, Shanghai 200240, PR China. [4]School of Mechanical Engineering, Purdue University, West Lafayette, IN 47907, USA. ✉e-mail: hua.bao@sjtu.edu.cn

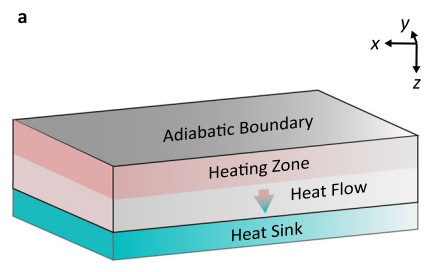

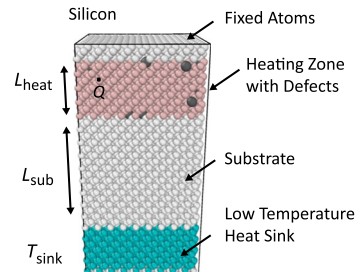

**Fig. 1 | The system of study. a** Schematic of the system of study. A thin film is sandwiched between an adiabatic boundary and a heat sink. Heat generation occurs in a heating zone near the adiabatic boundary. **b** Simulation cell adopted in molecular dynamics (MD) simulations. The substrate is pure Si and the heating zone is Si with Ge impurities occupying random sites.

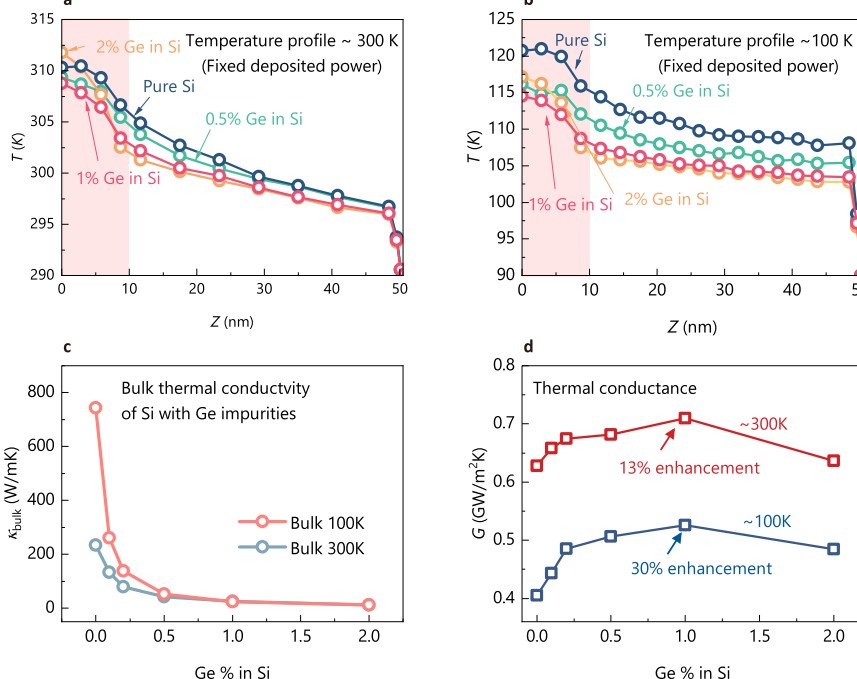

**Fig. 2 | MD simulation results. a** Temperature profiles along the $z$ direction around 300 K for different concentrations of Ge in the heating zone. **b** Temperature profiles along the $z$ direction around 100 K for different concentrations of Ge in the heating zone. **c** Bulk thermal conductivity of Si with Ge impurities. **d** Thermal conductance results for cases in (**a**, **b**).

reservoir[13]. Therefore, we model the heat generation in the heating zone using a Nose-Hoover chain thermostat and the fixed temperature heat sink using a Langevin thermostat. The length of the heating zone $L_{heat}$ and substrate $L_{sub}$ are set as 10 nm and 40 nm, respectively. The cases with other lengths will be discussed later. The length of the heat sink does not affect thermal transport and is set as 2 nm, which is sufficient to avoid nonphysical behavior[13,14]. We also try a system without fixed atoms by using periodic boundaries and find the negligible influence of fixed atoms (see Supplementary Note S1). Sufficient time is run to achieve steady-state. All the MD simulations are performed using the Large-scale Atomic/Molecular Massively Parallel Simulator (LAMMPS) package[15] (see Methods for more details).

We show the temperature profiles along the $z$ direction in Fig. 2a, b for different concentrations of Ge in the heating zone. We test the cases around two temperatures: 300 K (Fig. 2a) and 100 K (Fig. 2b), where the heat sink temperatures are fixed at 290 K and 90 K. For all cases, the deposited power in the heating zone is fixed at 5 eV/ps. Based on common understanding, heat dissipation of pure Si system should be better than the doped system due to the excessive scattering

induced by defects[2]. This can also be seen by our equilibrium MD (EMD) simulation results of bulk thermal conductivity of Si systems with Ge impurities: as shown in Fig. 2c, with 1% Ge, the bulk thermal conductivity is less than 10% of that of the pure Si. Hence, we would expect the temperature rise of the pure Si system should be the lowest with the fixed deposited power. Figure 2a, b, however, show unexpected trends: either around 300 K or 100 K, with the increase in defect concentration, the temperature rise in the heating zone first decreases and reaches the lowest at 1% concentration, and then increases with higher Ge concentration. This phenomenon shows that the heat dissipation of the system with 1% Ge is better than that of the pure Si system for the present problem. To quantify the performance of the heat dissipation, we define the thermal conductance $G = q_z/(T_{heat} - T_{sink})$ for this system, where $q_z$ is the heat flux along the $z$ direction in the substrate and $T_{heat}$ is the average temperature in the heating zone[16]. The definition of this thermal conductance is similar to the widely used thermal resistance in the heat dissipation of transistors, representing that, under the same heat generation, the lower the device temperature rise, the smaller the thermal resistance, and the

greater the thermal conductance[17–19]. As shown in Fig. 2d, for either 300 K or 100 K, the thermal conductance first increases and then decreases with increasing defect concentrations. The maximum enhancement in thermal conductance reaches 13% with 1% Ge at 300 K and reaches 30% with 1% Ge at 100 K. This significant increase in thermal conductance largely contradicts the drastic decrease of the bulk thermal conductivity as shown in Fig. 2c.

## Phonon transport mechanism

Because MD simulations study random atomic vibrations in real space, it is difficult to directly extract phonon transport physics[20]. In contrast, the phonon BTE based on the phonon gas model has been widely adopted for understanding thermal transport mechanisms[21–24]. In our previous studies, we have unified the two methods for thermal transport simulations[13]. Therefore, we adopt the phonon BTE to understand the mechanism behind thermal transport enhancement in this study.

At steady state, the energy-based phonon BTE with relaxation time approximation can be expressed as[25]:

$$\mathbf{v}_{\omega,p,\mathbf{s}} \cdot \nabla e_{\omega,p,\mathbf{s}} = \frac{e^0_{\omega,p} - e_{\omega,p,\mathbf{s}}}{\tau_{\omega,p}} + \frac{\dot{Q}_{\omega,p}}{4\pi}, \quad (1)$$

where $e$ is the phonon energy density[25] and $e^0$ is the corresponding equilibrium energy density. $\omega$, $p$, $\mathbf{s}$ represent the phonon frequency, polarization, and propagation direction, respectively. Here phonon energy density is defined as $e_{\omega,p,\mathbf{s}} = \hbar\omega D(\omega,p)(f_{\omega,p,\mathbf{s}} - f^0(T_{\text{sink}}))$, where $f$ is the phonon distribution function and $f^0$ is the equilibrium distribution function. The energy-based phonon BTE can be directly derived from the distribution function-based phonon BTE[25] (see Methods). The temperature of the system can be obtained from the summation of energy density over all phonon modes according to $T = \frac{1}{4\pi}\sum_p \int_{4\pi}\int_\omega e_{\omega,p,s}d\Omega d\omega / \sum_p \int C_{\omega,p}d\omega + T_{\text{sink}}$[25]. The heat flux of the system can be obtained from the summation of energy flux $\mathbf{e}_{\omega,p,\mathbf{s}} = e_{\omega,p,\mathbf{s}}\mathbf{s}$ over all phonon modes according to $\mathbf{q} = \int_{4\pi}\sum_p \int_\omega v_{\omega,p}\mathbf{e}_{\omega,p,\mathbf{s}}d\omega d\Omega$[25]. $\dot{Q}$ is the volumetric heat generation. $\mathbf{v}$, $\tau$, and $C$ are phonon group velocity, relaxation time, and heat capacity, respectively. The relaxation time in this study considers two scattering mechanisms: phonon–phonon scattering and phonon-defect scattering. According to the Matthiessen's rule, $1/\tau = 1/\tau_{\text{ph–ph}} + 1/\tau_{\text{ph–defect}}$[26,27]. Using the relaxation time implies that the phonon scattering processes are isotropic, i.e., no anisotropic scattering is induced artificially. We make the following simplifications in phonon BTE calculations. (i) We adopt the gray model for phonon properties, i.e., all phonons have the same properties including group velocity, heat capacity, and relaxation time. (ii) We assume that group velocity is the speed of sound (5500 m/s for Tersoff Si)[28,29]. Heat capacity is the total heat capacity 2.06 $J/m^3K$[11]. (iii) In the heating zone, the relaxation time of the phonon–phonon scattering is obtained according to $\tau_{\text{ph–ph}} = 3k_{\text{bulk}}/Cv^2$, where $k_{\text{bulk}}$ is the bulk thermal conductivity. The values of phonon–phonon relaxation time are $3.93 \times 10^{-11}$ s and $1.30 \times 10^{-11}$ s for 100 K and 300 K, respectively. The relaxation time of phonon-defect scattering varies in a range from $10^{-9}$ to $10^{-12}$ s. (iv) All scatterings in the substrate are neglected. These simplifications can help better reveal the underlying mechanisms. The rigorous calculations that consider the mode-level phonon properties and defect scattering will be discussed later. Based on the assumptions above, the phonon BTE can be numerically solved to obtain the temperature profile and heat flux (see Methods for numerical details).

The computational domain of the phonon BTE is shown in Fig. 3a. The upper boundary is set as the specularly reflecting boundary (see Methods), a type of adiabatic boundary condition in BTE, which has been proven to be equivalent to the fixed layer of atoms in the MD

simulation[13]. Uniform heat generation is set in a heating zone with the length of 10 nm next to the specularly reflecting boundary. The volumetric heat generation rate is fixed at $2 \times 10^{17} W/m^3$ for all cases; thus the heat flux in z directions is fixed at $2 \times 10^9 W/m^2$ according to the energy balance $\int \dot{Q}dV = \int \mathbf{q} \cdot \mathbf{n}dA$. The substrate has a length of $L_{\text{sub}} = 40$ nm. The bottom boundary is set as the thermalizing boundary condition, which is equivalent to the Langevin thermostat[13]. The thermalizing boundary condition emits phonons in their equilibrium state with a fixed temperature and absorbs all injected phonons (Fig. 3a). Such a computational domain is equivalent to the MD simulation domain as Fig. 1b. The thermal conductance results obtained using the phonon BTE are shown in Fig. 3b. In phonon BTE calculations, we find similar trends of thermal conductance as in MD simulations (Fig. 2d): the thermal conductance first increases and then decreases with decreasing phonon-defect relaxation time (increasing defect concentration), at both 100 K and 300 K.

Both MD and BTE show that introducing defect scattering can lead to enhanced thermal transport. As will be shown later, with consideration of the mode-level phonon properties, the two methods can be quantitatively matched, providing robust evidence for the accuracy of our calculations and supporting the credibility of our findings. To uncover the mechanism of this intriguing phenomenon, we first analyze how the temperature distribution varies with the phonon-defect relaxation time. Figure 4a shows the temperature distribution at 100 K for pure Si and systems with phonon-defect relaxation time of $5 \times 10^{-12}$ s and $10^{-12}$ s at 100 K. For these three cases, there is a temperature drop inside the heating zone and another temperature drop from the heating zone-substrate interface to the bottom boundary. Note that the temperature inside the substrate is constant since the scattering inside the substrate is neglected[30]. The temperature drop from the heating zone-substrate interface to the bottom boundary is equal to the temperature jump at the boundary in present cases. We divide the total temperature drop into two parts: the temperature drop inside the heating zone $\Delta T_{\text{heat}}$ and the temperature drop from the heating zone-substrate interface to the bottom boundary $\Delta T_{\text{sub}}$, i.e., temperature jump at the boundary (Fig. 4a). Figure 4b shows how these two temperature drops vary with phonon-defect relaxation time. With decreasing phonon-defect relaxation time, the temperature drop inside the heating zone $\Delta T_{\text{heat}}$ increases but $\Delta T_{\text{sub}}$ decreases. Therefore, the overall thermal conductance $G$ ($(G = q_z/(\overline{\Delta T_{\text{heat}}} + \Delta T_{\text{sub}})$, where $\overline{\Delta T_{\text{heat}}}$ denotes the average temperature drop inside the heating zone relative to the heating zone-substrate interface, with the calculation expressed as $T_{heat} - T_{cold} - \Delta T_{\text{sub}}$) first increases due to the decreasing $\Delta T_{\text{sub}}$ when phonon-defect relaxation time is large and then decreases due to the increasing $\Delta T_{\text{heat}}$ when phonon-defect relaxation time becomes too small. The increase of $\Delta T_{\text{heat}}$ is consistent with the previous understanding that the scattering impedes phonon transport[30,31]. The decreasing $\Delta T_{\text{sub}}$ leads to the unexpected increase of the thermal conductance. To understand the decreasing $\Delta T_{\text{sub}}$, we further examine the directional phonon energy flux.

We calculate the directional phonon energy flux distribution for pure Si and systems with phonon-defect relaxation time of $5 \times 10^{-12}$ s and $10^{-12}$ s at 100 K, as shown in Fig. 4c. One can see that for pure Si systems, phonons propagating in different directions exhibit large nonequilibrium. When the phonon defect scattering increases, the directional phonon nonequilibrium significantly reduces, leading to a more isotropic phonon energy flux distribution. To understand why this directional nonequilibrium exists and how it affects thermal transport, the transport mechanism is illustrated in Fig. 4d, e. When the scattering is rare (i.e., in the ballistic regime) inside the uniform volumetric heating zone (Fig. 4d), the phonon mode propagating oblique to the z direction (mode 2) travels a much longer distance in the heating zone than the mode propagating along the z direction (mode 1), and thus receives a much larger amount of energy. Therefore, the phonons propagating in different directions exhibit large directional nonequilibrium (Fig. 4d).

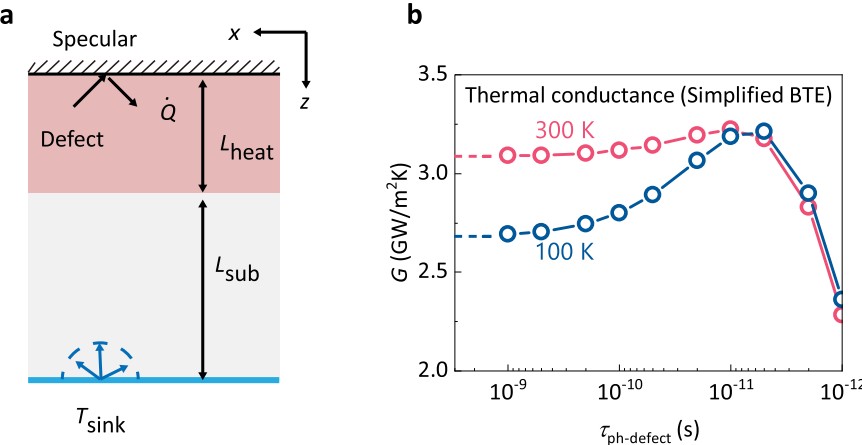

**Fig. 3 | Calculated results from the simplified phonon BTE. a** The computational domain of the phonon BTE calculations. **b** Thermal conductance results obtained using simplified phonon BTE calculations.

Based on the definition of temperature under such directional none-quilibrium, the over-populated mode 2 will have a larger contribution to the local temperature than mode 1. On the other hand, the local heat flux is only contributed by the energy flux projected along $z$ directions, hence mode 2 only has a small contribution to the local heat flux (Fig. 4d). If sufficient scattering is present in the heating zone (i.e., in the diffusive regime), scattering can randomly redirect phonons and restore the equilibrium directional distribution, as depicted in Fig. 4e. Now we compare the cases of Fig. 4d, e. If the heat fluxes in both cases are the same, the case in Fig. 4d will have a higher temperature than the one in Fig. 4e. This directional nonequilibrium phonon transport explains why the temperature drop in the substrate $\Delta T_{sub}$ decreases with decreasing phonon-defect relaxation time. In other words, the defect-free heating zone overpopulates the oblique-propagating phonons and leads to lower thermal conductance than the equilibrium situation. Introducing defects redirects phonons randomly to restore directional equilibrium and thereby enhances thermal conductance. At 300 K, due to the smaller phonon–phonon relaxation time[32], the phonon transport for pure Si is more diffusive than that at 100 K, and the directional phonon nonequilibrium is smaller for pure Si (see Supplementary Note S2). As a result, the thermal conductance enhancement is smaller at 300 K than that of 100 K (Fig. 3b). Note that ref. 33. recently utilized defects to redirect phonons and assist phonon transport through kinked nano-wires, which differs in principle from this study that focused on redu-cing directional nonequilibrium through defect scattering. Meanwhile, previous discussions of phonon local nonequilibrium focused mainly on spectral nonequilibrium, i.e., different phonon modes are in nonequilibrium[29,34–36]. This work introduces a different type of local nonequilibrium, i.e., directional nonequilibrium, to the family of phonon local nonequilibrium transport.

## Mode-level phonon BTE calculations
From the simplified BTE analysis, we reveal that the enhancement of thermal conductance is due to the manipulation of directional none-quilibrium by defect scattering. To further validate the simplified BTE analysis, we conduct rigorous mode-level phonon BTE calculations without the simplifications. The input phonon properties, including the group velocity, heat capacity, and relaxation time, are obtained using the standard anharmonic lattice dynamics considering up to three-phonon scattering[37]. To consider the defect scattering by Ge atoms, we adopt the Tamura model[38], which considers scattering caused by mass difference and is proven to be quite accurate[4,26]. All the lattice dynamics calculations are performed on the ALAMODE package[39]. To quantita-tively compare the phonon BTE and MD results, we first extract phonon properties from Tersoff potential[12] and assume the phonon population

to be classical. Mode-level heat generation is distributed over the pho-non mode according to its heat capacity[13]. Thermal conductance obtained from this mode-level phonon BTE calculation is compared with MD results and shown in Fig. 5a. It can be clearly seen that with rigorous calculations, quantitative agreement in conductance can be achieved between BTE and MD results (differences are less than 10%). Since the MD simulation adopts the empirical Tersoff potentials[12] and takes the classical distribution of the phonon population[11], it is still dif-ferent from the real situation. We further obtain phonon properties from first-principles calculations and consider quantum phonon popu-lation (Bose-Einstein distribution). The obtained thermal conductance results for Si with different Ge impurity concentrations are shown in Fig. 5b. One can see that the overall trend of thermal conductance with the Ge impurity concentrations is the same as that in MD simulations and simplified BTE calculations. Therefore, we can conclude that defect scattering in this case reduces the unfavored directional none-quilibrium, hence leads to enhanced thermal transport in the system in Fig. 1 ($L_{heat}$ = 10 nm and $L_{sub}$ = 40 nm).

## Influence factor
In the above discussions, we focus on the Si systems with fixed temperatures and sizes. To further examine the generality of direc-tional phonon nonequilibrium, we study systems with different temperatures, sizes, and materials. We first investigate the Si systems with fixed sizes at four temperatures (400 K, 300 K, 100 K, and 50 K) and observe the thermal conductance enhancement at all tempera-tures (Fig. 5b). Especially, the maximum enhancement of thermal conductance reaches 61% for 50 K. The enhancement at low tem-peratures is important for cryogenic transistors in quantum com-putation and radio astronomy[23,40]. Figure 5c shows the thermal conductance for three different lengths of substrates (10 nm, 100 nm and 1 $\mu$m) with a fixed length of heating zone (10 nm) at 100 K. For all these lengths, thermal conductance enhancement by introducing defects is observed. The maximum enhancement of thermal con-ductance reaches 75% with a 100 nm substrate. In Fig. 5d, we fix the length of substrate (1 $\mu$m) and compute the thermal conductance at 100 K for three lengths of heating zone (10 nm, 100 nm and 1 $\mu$m). With the increasing lengths of the heating zone, the increase in thermal conductance gradually decreases and eventually disappears. The reason is that with increasing size, the phonon transport in the heating zone becomes more diffusive due to the intrinsic phonon–phonon scattering and the directional phonon none-quilibrium gradually reduces. We also investigate the thermal con-ductance of SiC and GaN systems, which serve as channel materials for power transistors[41,42], as illustrated in Fig. 6. In SiC systems, we

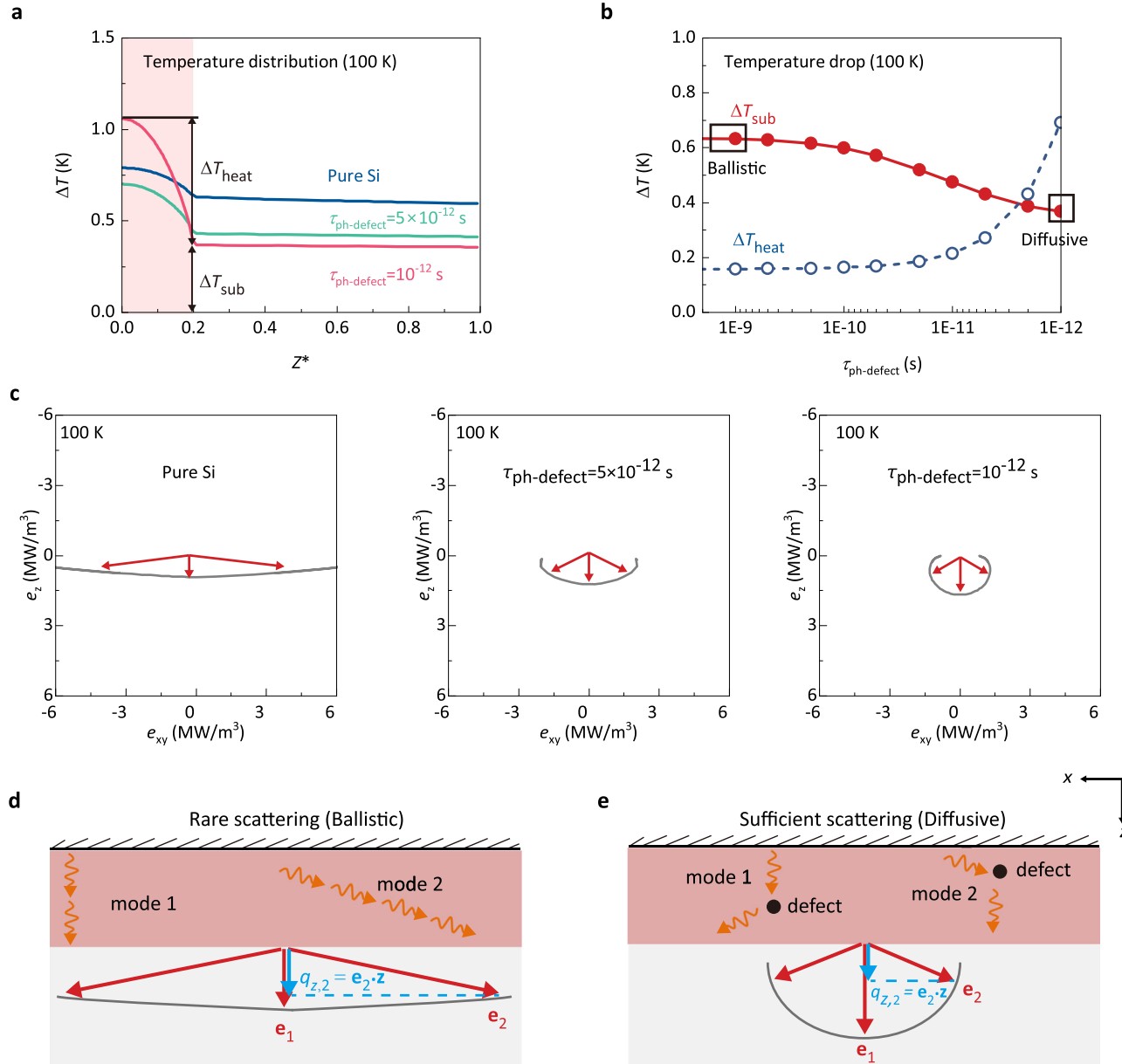

**Fig. 4 | Phonon transport mechanism. a** Temperature distribution at 100 K obtained from simplified phonon BTE calculations. We divide the temperature drop into two parts: the temperature drop inside the heating zone $\Delta T_{heat}$ and the temperature drop from the heating-substrate interface to the bottom $\Delta T_{sub}$, i.e., temperature drop in the substrate. **b** The temperature drop inside the heating zone $\Delta T_{heat}$ and temperature drop in the substrate $\Delta T_{sub}$ for different phonon-defect relaxation time. **c** Calculated directional phonon energy flux in the substrate. The $x$-axis represents the projection of the energy flux on the $x$-$y$ plane. The $y$-axis represents the projection of the energy flux on the $z$ direction. **d** Phonon transport mechanism in the ballistic regime. When the scattering is rare (i.e., in the ballistic

regime) inside the uniform volumetric heating zone, the phonon mode propagating oblique to the $z$ direction (mode 2) travels a much longer distance in the heating zone than the mode propagating along the $z$ direction (mode 1), and thus receives a much larger amount of energy. Therefore, the phonons propagating in different directions exhibit large directional nonequilibrium ($e_2$ is much larger than $e_1$). the local heat flux is only contributed by the energy flux projected along $z$ directions (blue arrow). **e** Phonon transport mechanism in the diffusive regime. If sufficient scattering is present in the heating zone (i.e., in the diffusive regime), scattering can randomly redirect phonons and restore the equilibrium directional distribution.

introduce defect scattering by randomly replacing C atoms with $^{14}$C atoms. For GaN systems, defect scattering is induced by randomly replacing Ga atoms with $^{71}$Ga atoms. Additionally, we explore the impact of $^{24}$Mg impurities in GaN and $^{10}$B impurities in SiC, as detailed in Supplementary Notes S5 and S6. Phonon properties are obtained from first-principles calculations (see Supplementary Notes S5 and S6). Our investigation encompasses varying temperatures (Fig. 6a, d), substrate lengths (Fig. 6b, e), and heating zone lengths (Fig. 6c, f). We observe thermal conductance enhancement by introducing defect scattering across different temperatures and sizes. As such, in a wide range of temperatures, sizes, and materials,

directional nonequilibrium exists and can be mitigated by defect scattering to enhance thermal transport. The only requirement for the existence of this directional nonequilibrium is that the phonon transport in the heating zone is in the ballistic regime, i.e., the length of the heating zone is smaller than the phonon mean free path.

Previous studies of phonon local nonequilibrium focused mainly on spectral nonequilibrium[29,34–36]. Some studies found that in transistors or Raman measurements, optical phonons are over-populated which leads to lower thermal conductance[34–36]. Our analysis so far has assumed that mode-level heat generation is proportional to the modal heat capacity (spectral equilibrium). However, we also

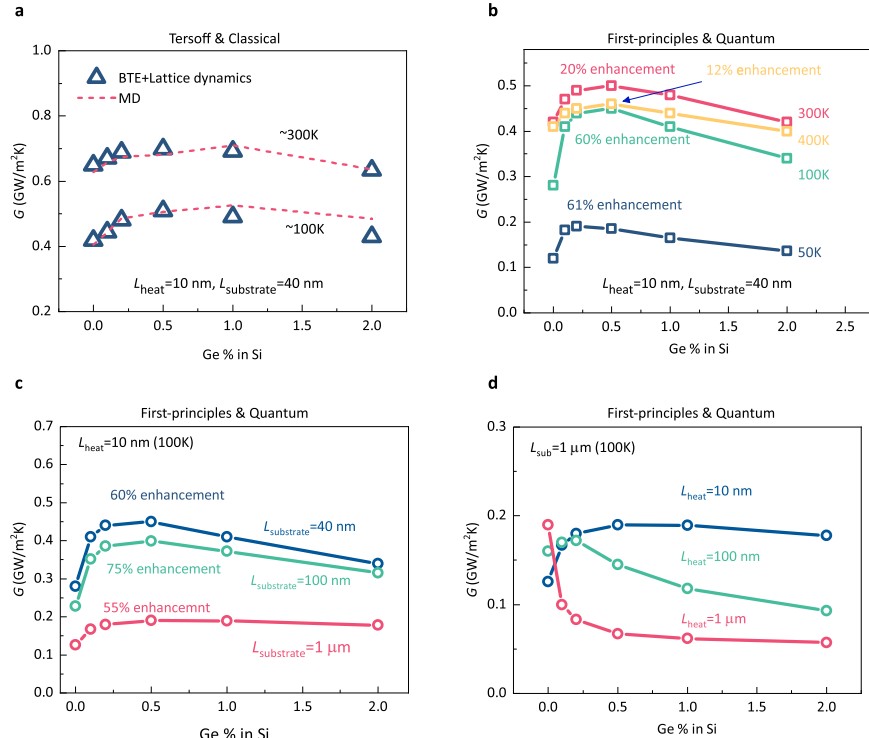

**Fig. 5 | Thermal conductance results obtained from mode-level phonon BTE calculations. a** Thermal conductance with Tersoff potential and classical population. **b** Thermal conductance at different temperatures with first-principles phonon properties. **c** Thermal conductance at 100 K with a fixed length of heating zone (10 nm) and different lengths of substrate (first-principles phonon properties). **d** Thermal conductance at 100 K with a fixed length of substrate (1 $\mu$ m) and different lengths of heating zone (first-principles phonon properties).

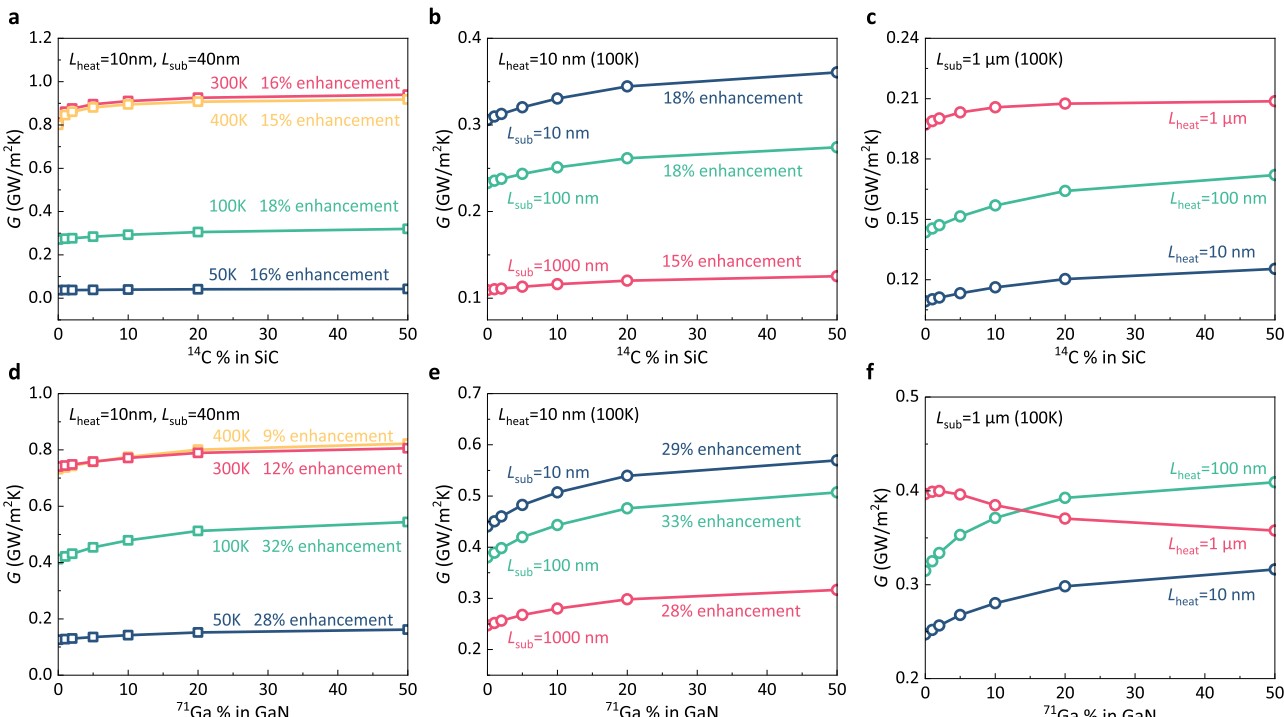

**Fig. 6 | Thermal conductance of 4H-SiC and wurtzite GaN systems. a** Thermal conductance of 4H-SiC systems at different temperatures. **b** Thermal conductance of 4H-SiC systems at 100 K with a fixed length of the heating zone (10 nm) and different lengths of the substrate. **c** Thermal conductance of 4H-SiC systems at 100 K with a fixed length of the substrate and different lengths of the heating zone. **d** Thermal conductance of wurtzite GaN systems at different temperatures. **e** Thermal conductance of wurtzite GaN systems at 100 K with a fixed length of the heating zone (10 nm) and different lengths of the substrate. **f** Thermal conductance of wurtzite GaN systems at 100 K with a fixed length of the substrate and different lengths of the heating zone.

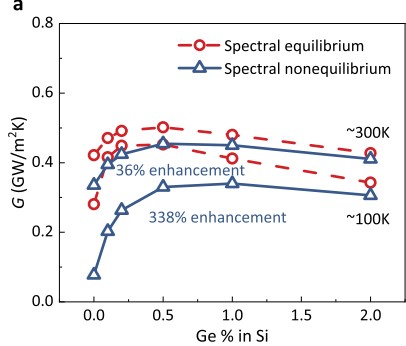
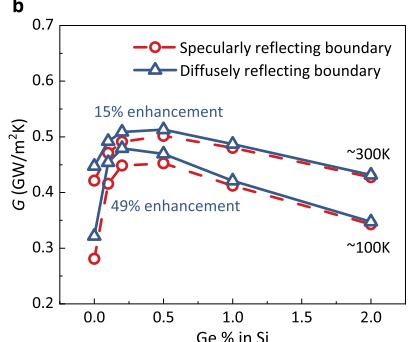
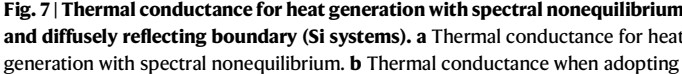

**Fig. 7 | Thermal conductance for heat generation with spectral nonequilibrium and diffusely reflecting boundary (Si systems). a** Thermal conductance for heat generation with spectral nonequilibrium. **b** Thermal conductance when adopting the diffusely reflecting boundary. Si systems with 10 nm heating zone and 40 nm substrate are studied.

investigate the thermal conductance with considering the over-population of optical phonons (spectral nonequilibrium). We conduct a rigorous electron–phonon coupling calculation for Si to determine the mode-level heat generation (see Supplementary Note S7). As shown in Fig. 7a, compared with cases for heat generation with spectral equilibrium, we actually find more significant thermal conductance enhancement by introducing defect scattering for heat generation with spectral nonequilibrium. At 100 K, the maximum thermal conductance enhancement reaches 338%. The underlying mechanism is that defect scattering not only manipulates the directional nonequilibrium but also the spectral nonequilibrium (see Supplementary Note S7).

In all the aforementioned discussions, we consistently applied a specular reflection boundary condition to the upper boundary across all systems. To show the robustness of our findings, we also explored systems with diffuse reflection boundary conditions (see Methods). As illustrated in Fig. 7b, a decrease in thermal conductance enhancement is observed when compared to cases with specular boundaries. This is expected as diffuse boundary conditions are similar to defect scattering and tend to randomize the phonon directions, thereby contributing to the reduction of directional nonequilibrium. It is noteworthy that both specular and diffuse reflection boundaries can be experimentally implemented[43,44], and the boundary in reality is most likely partially specular and partially diffuse. Consequently, our research offers valuable insights into optimizing thermal conductance enhancement through impurity scattering in experimental settings, with a preference for the utilization of more specular boundaries.

We have noticed another work by ref. 45. published during the peer review process of this study, which doped oxygen atoms in van der Waals crystal TiS$_3$ nanoribbons to induce lattice contraction and increase Young's modulus, leading to enhanced thermal conductivity. Distinct from inducing defects to migrate directional nonequilibrium in this study that reveals a general mechanism to enhance thermal transport, the strategy provided by ref. 45 is available for limited material systems. Nevertheless, the potential coexistence of these effects provides further viable approaches for manipulating heat transfer at the micro- and nanoscale. The observed phenomenon in this study also holds promise for experimental validation. Our system has real-world counterparts, where the volumetric heating method aligns with optical and electrical heating in experiments[36,41]. The corresponding heat sink corresponds to the thermal reservoir in the heat bridge method[46,47]. Our discussions on the influencing factors also provide guidance on maximizing the enhancement of thermal conductance through defect scattering in experiments (at lower temperatures, with smaller sizes, and with a specular upper surface).

In summary, we demonstrate that defect scattering could enhance phonon transport at the nanoscale and reveal the underlying mechanism of this phenomenon. To achieve a phonon transport enhancement by introducing defect scattering, the only requirement is that the phonon transport in the heating zone is in the ballistic regime, i.e., the length of the heating zone is smaller than the phonon mean free path. Our further and comprehensive analysis shows that the defect-free volumetric heating zone overpopulates oblique-propagating phonons, while introducing defects would redirect phonons randomly to restore directional equilibrium, thereby enhancing thermal conductance. Such a mechanism exists in a wide range of temperatures (from cryogenic temperature to above room temperature), materials (Si, SiC, GaN), and sizes (up to microns). For the cases we studied, the enhancement in thermal conduction via introducing defect scattering can be up to 75%. This work introduces a different type of local nonequilibrium, i.e., directional nonequilibrium, to the family of phonon local nonequilibrium transport, and extends our current strategies for enhancing thermal transport. Moreover, introducing defect scattering into heating region can be a simple and effective strategy to further overcome heat dissipation bottlenecks for a variety of transistors including computational architectures, power transistors, and cryogenic transistors.

## Methods
### Molecular dynamics simulations
All the MD simulations are performed using the Large-scale Atomic/Molecular Massively Parallel Simulator (LAMMPS) package[15]. In the simulation, the time step is set as 1 fs for 300 K and 0.1 fs for 100 K, respectively. The whole system is first relaxed under the NPT (constant mass, pressure, and temperature) ensemble for 3,000,000 steps, after which we build the heating zone and heat sink (the Nose-Hoover chain thermostat and the Langevin thermostat). The simulations are then run for 20,000,000 steps, while the steady state can be well achieved with the first 5,000,000 steps for all our simulated systems. We use the data recorded within the last 15,000,000 steps to determine the temperature profile and the heat flux. The heat flux values in the substrate are obtained by energy conservation[48]. The deposited power in the heating zone is fixed as 5 eV/ps. Since the Nose-Hoover thermostat can only control average temperature, we actually modify the target average temperature of the Nose-Hoover thermostat for different cases to ensure the fixed deposited power. In both the Langevin and Nose-Hoover chain thermostats, the re-scaling time constant is 1/100 of the time step, as recommended by a previous work[14].

### Phonon Boltzmann transport equation
The phonon Boltzmann transport equation is expressed as[30]:

$$\mathbf{v}_{\omega,p,\mathbf{s}} \cdot \nabla f_{\omega,p,\mathbf{s}} = \frac{f^0 - f_{\omega,p,\mathbf{s}}}{\tau_{\omega,p}} + \dot{s}_{\omega,p,\mathbf{s}}, \tag{2}$$

where $f$ is the phonon distribution function, $f^0$ is the equilibrium distribution, and $\dot{s}$ is the source term. By defining a phonon energy density as $e_{\omega,p,s} = \hbar\omega D(\omega,p)(f_{\omega,p,s} - f^0(T_{\sin k}))$, the energy-based phonon BTE can be derived as[25]:

$$\mathbf{v}_{\omega,p,\mathbf{s}} \cdot \nabla e_{\omega,p,\mathbf{s}} = \frac{e^0_{\omega,p} - e_{\omega,p,\mathbf{s}}}{\tau_{\omega,p}} + \frac{\dot{Q}_{\omega,p}}{4\pi}. \qquad (3)$$

$e^0$ is the corresponding equilibrium energy density, which is corresponding energy density of the equilibrium distribution $f^0$, which is obtained according to energy conservation[25]:

$$e^0_{\omega,p} = \frac{C_{\omega,p}}{4\pi} \sum_{p'} \int_{4\pi} \int_{\omega'} \frac{e_{\omega',p',\mathbf{s}}}{\tau_{\omega',p}} d\Omega d\omega' \Big/ \sum_{p'} \int \frac{C_{\omega',p'}}{\tau_{\omega',p'}} d\omega'. \qquad (4)$$

The temperature of the system is obtained according to (consistent with the temperature definition in MD simulation)[13]

$$T = \frac{1}{4\pi} \sum_p \int_{4\pi} \int_\omega e_{\omega,p,\mathbf{s}} d\Omega d\omega \Big/ \sum_p \int C_{\omega,p} d\omega + T_{\text{sink}}. \qquad (5)$$

The heat flux of the system is obtained according to

$$\mathbf{q} = \int_{4\pi} \sum_p \int_\omega \mathbf{v}_{\omega,p} e_{\omega,p,\mathbf{s}} d\omega d\Omega. \qquad (6)$$

The specularly reflecting boundary condition for phonons with the incident direction $\mathbf{s}_r$ and the reflected direction $\mathbf{s}$ is specified by[25,49]

$$e_{\omega,p,\mathbf{s}} = e_{\omega,p,\mathbf{s}_r} (\mathbf{s} \cdot \mathbf{n} < 0). \qquad (7)$$

On the other hand, phonons reflected from diffusely reflecting boundaries have the same energy in each direction, i.e.,

$$e_{\omega,p,\mathbf{s}} = \frac{1}{\pi} \int_{\mathbf{s}' \cdot \mathbf{n} > 0} e_{\omega,p,\mathbf{s}'} \mathbf{s}' \cdot \mathbf{n} d\Omega, \qquad (8)$$

where $\mathbf{n}$ is the exterior normal unit vector of the boundary.

The thermalizing boundary condition is expressed as[13,25,49]

$$e_{\omega,p,\mathbf{s}} = \frac{C_{\omega,p}}{4\pi}(T_B - T_{\sin k})(\mathbf{s} \cdot \mathbf{n} < 0), \qquad (9)$$

where $\mathbf{n}$ is the exterior normal unit vector of the boundary. $T_B$ is the temperature of the thermalizing boundary condition.

In-house code based on the discrete-ordinate method and finite volume method[49] is adopted to solve the phonon BTE. The propagation direction and spatial domain is discretized, and the phonon BTE is transformed into a set of algebraic equations. These algebraic equations are solved iteratively. After a convergence test, we sample 512 propagating directions and 1000 spatial meshes for all cases.

## Lattice dynamics calculations

To perform the mode-level phonon BTE calculations, the phonon properties including the group velocity, heat capacity, and relaxation time are needed as input information. These parameters are obtained using the standard anharmonic lattice dynamics approach considering up to three-phonon scattering[37,39]. To consider the defect scattering by Ge atoms, we adopt the Tamura model[38]. All the lattice dynamics calculations are performed on the ALAMODE package[39].

When comparing the MD and phonon BTE results, the phonon properties are extracted from the same system as the MD simulation: Si crystal with Tersoff potential[12]. The phonon population is assumed to be classical. We compute atomic forces for displaced configurations using the LAMMPS package[15] and extract harmonic and third-order anharmonic interatomic force constants by fitting the relation between atomic forces and the displacements. A supercell of $4 \times 4 \times 4$ and the second nearest atom neighbor is considered to obtain the third-order anharmonic interatomic force constants. The thermal conductivity of Si is calculated based on the single-mode relaxation time approximation method. We use $60 \times 60 \times 60$ q-points to sample the Brillouin zone. The obtained values of classical thermal conductivity at 100 K and 300 K are 817 W/m·K and 272 W/m·K, which has around 10% difference with the EMD results (Fig. 1c) may due to the ignorance of higher-order scattering process or the single-mode relaxation time approximation[11].

To consider more realistic situations, we extract phonon properties from first-principles calculations with the quantum phonon population (Bose-Einstein distribution). We carried out the first-principles calculations using the Quantum ESPRESSO package[50] to calculate the atomic forces and extract harmonic and third-order anharmonic interatomic force constants by fitting the relation between atomic forces and the displacements. A supercell of $4 \times 4 \times 4$ and the fourth nearest atom neighbor is considered to obtain the third-order anharmonic interatomic force constants. The thermal conductivity of Si is calculated based on the single-mode relaxation time approximation method. We use $60 \times 60 \times 60$ q-points for all temperatures to sample the Brillouin zone. The bulk thermal conductivity of Si for different temperatures is provided in Supplementary Note S3.

In the BTE solver, we cannot directly sample phonon modes from lattice dynamics calculations due to the huge computational cost. After a convergence test, we divide the phonon frequency into 200 bins and average the properties of phonons in each bin according to[49]:

$$C_n = \sum_{\omega_n}^{\omega_{n+1}} C_{\omega,p}, \nu_n = \frac{\sum_{\omega_n}^{\omega_{n+1}} C_{\omega,p} \nu_{\omega,p}}{\sum_{\omega_n}^{\omega_{n+1}} C_{\omega,p}}, \tau_n = \frac{\sum_{\omega_n}^{\omega_{n+1}} C_{\omega,p} \nu^2_{\omega,p} \tau_{\omega,p}}{\nu_n \sum_{\omega_n}^{\omega_{n+1}} C_{\omega,p} \nu_{\omega,p}}. \qquad (10)$$

## Data availability

The authors declare that the data supporting the findings of this study are available in the main article, Supplementary Information and Source data file. Source data are provided with this paper.

## Code availability

The authors declare that the code supporting the findings of this study is available from the corresponding author upon request.

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

## Acknowledgements

Y.H., J.X., and H.B. acknowledge the support by the National Natural Science Foundation of China (Grant No. 52122606, H.B.). We would also like to thank Dr. Xiaokun Gu from Shanghai Jiao Tong University for valuable discussion on lattice dynamics calculations. The computations in this paper were run on the $\pi$ 2.0 cluster supported by the Center for High Performance Computing at Shanghai Jiao Tong University.

## Author contributions

H.B. and Y.H. conceived and designed the research, H.B. supervised the research. Y.H. and J.X. performed numerical simulation, analyzed the data, and prepared the figures and manuscript with discussions with

X.R. X.R. revised the paper. All the authors contributed to the data analysis and the preparation of the manuscript.

## Competing interests

The authors declare no competing interests.
