## [Peer Review File · Nature Communications]

Defect scattering can lead to enhanced phonon transport at nanoscaleREVIEWER COMMENTS

Reviewer #1 (Remarks to the Author):

This manuscript demonstrates a novel phenomenon that defect scattering in the heating zone can counterintuitively enhance thermal transport at nanoscale, which breaks the common understanding that defect scattering always impedes thermal transport.

This novel finding is calculated by using both molecular dynamics and phonon Boltzmann transport equations, which seems to be rigorously done and the quantitative agreements comparing with different approaches make the results solid and convincing. The counterintuitive phenomenon is well-explained by the phonon direction nonequilibrium mechanism. Moreover, the widespread existence of this phenomenon is reported across different temperatures, materials, and sizes brings a possible solution for further overcoming the heat dissipation bottleneck in electronics. For these reasons, I recommend this intriguing paper for publication in Nature Communications after the following issues are addressed.

(i) The title needs to be modified to "Defect scattering can lead to enhanced phonon transport at nanoscale", which would better convey the contribution of this study.

(ii) The results regarding various materials and spectral nonequilibrium in supplementary materials are significant and can be moved to the main text if space is not limited.

(iii) The results with regard to the over-population of optical phonons are interesting, as the inclusion of defect scattering leads to a more significant increase in thermal conductance. This is based on an assumption that optical phonons obtain all energy. Recent electron-phonon coupling calculations show that some acoustic phonons can still obtain energy from electrons. It would be better if the authors discuss how significant this effect can be if the optical phonons obtain less energy (for example, according to rigorous electron-phonon coupling calculation).

(iv) There is a mistake in Fig. 4c where the center of the red arrow does not align with the gray line, which should be corrected.

(v) In Wurtzite GaN and 4H-SiC systems, inducing defect scattering seems to monotonously increase the thermal conductance. If one further increases the defect scattering in Wurtzite GaN and 4H-SiC systems, will the thermal conductance decrease like that in the silicon system? It is better if some discussions are provided.

Reviewer #2 (Remarks to the Author):

This manuscript reports on a study of the effects of defect scattering in the heating zone on the calculated thermal conductance using molecular dynamics simulation and numerically solving the Boltzmann transport equation. Contrary to the common expectation of enhanced resistance from defects, the results here show that defects in the heat zone could increase the derived thermal conductance. The authors attributed the enhanced thermal conductance to the defect scattering induced spatial redistribution of phonon energy and introduced the concept of directional non-equilibrium.

The authors have conducted extensive modeling to prove their arguments and the referee feels that the disclosed underlying physics is interesting. However, the referee is not convinced that the observed phenomena are not due to numerical artifacts in the simulations. Therefore, the manuscript has to be reconsidered after the authors address the following concerns.

1. The authors claimed that “since both MD and BTE can show that introducing defect scattering can lead to enhanced thermal transport, it is clearly not caused by simulation artifacts.” The referee disagrees with this argument because in the BTE calculation, the boundary conditions are set to match the transport mechanism in the MD. As such, it is not a surprise that the MD and BTE yielded similar trends.

2. The referee has the concern that the effect is actually induced by simulation artifacts, i.e., the way that heat is added to the heating zone. In fact, the authors stated “When the scattering is rare (i.e., in the ballistic regime) inside the uniform volumetric heating zone (Fig. 4d), the phonon mode propagating oblique to the z direction (mode 2) travels a much longer distance in the heating zone than the mode propagating along the z direction (mode 1), and thus receives a much larger amount of energy.” The referee feels that this explanation exactly suggests that the effect is indeed due to how energy is added to the heating zone.

3. The authors set the reflection at the upper boundary as specular, which helps maintain the direction of phonons from the normal direction of the boundary. Based on the referee’s understanding of the manuscript, if the upper boundary is set as fully diffuse reflection/scattering, the effect will be significantly diminished, which would mean that indeed the phenomenon is a simulation artifact. Similarly, in MD, if the interface between the fixed atoms and the heating zone is set to be rough, the effects should be less significant.

4. The definition of $\Delta T_{\text{heat_bar}}$ in the first line of Page 9 is confusing.

5. In the derivation of thermal conductance from MD, the authors adopted an equation that includes the effects of the heat source and sink. Therefore, the derived conductance includes the boundary effect. What if the authors adopt the conductance as the ratio of heat flux over the temperature gradient in the substrate?

6. Since there is no scattering in the substrate in the BTE modeling, the definition of ΔT_{sub} is confusing. It is a temperature jump at the lower boundary instead of a temperature drop across the substrate.

7. Under special conditions, defects can actually lead to enhanced thermal conductance as demonstrated by quite a few experiments. The authors should discuss their results with respect to those experiment results and provide perspectives on the different transport mechanisms.

Reviewer #3 (Remarks to the Author):

In this paper, the authors show that the defect-free heating zone overpopulates oblique-propagating phonons, while introducing defects would redirect phonons randomly to restore directional equilibrium. They demonstrate that defect scattering can enable such thermal transport enhancement in a wide range of temperatures, and offer an unconventional strategy for enhancing thermal transport. The results are interesting and important.

There are some issues for authors to address:

- i): A cross-sectional area of 8×8 is not large, whether there is an effect of phonon-boundary scattering on thermal conductivity needs to be further demonstrated.
- ii): How do phonons with different vibrational frequencies contribute differently to the thermal conductivity? The PDOS of phonons with different doping concentrations are useful to complement the phonon transport mechanism.
- iii): In Fig. 5c, why does the enhancement of G first increase and then decrease as the substrate length increases?

The authors sincerely appreciate the constructive comments concerning our manuscript from all reviewers. Based on these comments, we have made corresponding revisions carefully that have been highlighted in the revised manuscript and supplementary materials using blue font. We believe the revised version has adequately addressed all of the comments. The point-by-point responses to the comments are listed below.

Response to Reviewer #1:

This manuscript demonstrates a novel phenomenon that defect scattering in the heating zone can counterintuitively enhance thermal transport at nanoscale, which breaks the common understanding that defect scattering always impedes thermal transport.

This novel finding is calculated by using both molecular dynamics and phonon Boltzmann transport equations, which seems to be rigorously done and the quantitative agreements comparing with different approaches make the results solid and convincing. The counterintuitive phenomenon is well-explained by the phonon direction nonequilibrium mechanism. Moreover, the widespread existence of this phenomenon is reported across different temperatures, materials, and sizes brings a possible solution for further overcoming the heat dissipation bottleneck in electronics. For these reasons, I recommend this intriguing paper for publication in Nature Communications after the following issues are addressed.

Response: We are grateful to the reviewer for the positive evaluation.

(i) The title needs to be modified to "Defect scattering can lead to enhanced phonon transport at nanoscale", which would better convey the contribution of this study.

Response: Thanks for the suggestion. We agree that the new title would better convey the contribution of this study. We have modified the title of this manuscript to "Defect scattering can lead to enhanced phonon transport at nanoscale".

(ii) The results regarding various materials and spectral nonequilibrium in

supplementary materials are significant and can be moved to the main text if space is not limited.

Response: Thanks for the suggestion. We have modified and moved the results regarding various materials and spectral nonequilibrium to the updated manuscript (Pages 14-16):

“We also investigate the thermal conductance of SiC and GaN systems, which serve as channel materials for power transistors [40,41], as illustrated in Fig. 6. In SiC systems, we introduce defect scattering by randomly replacing C atoms with ^{14}C atoms. For GaN systems, defect scattering is induced by randomly replacing Ga atoms with ^{71}Ga atoms. Additionally, we explore the impact of ^{24}Mg impurities in GaN and ^{10}B impurities in SiC, as detailed in Supplementary Notes S5 and S6. Phonon properties are obtained from first-principles calculations (see Supplementary Notes S5 and S6). Our investigation encompasses varying temperatures (Fig. 6a and d), substrate lengths (Fig. 6b and e), and heating zone lengths (Fig. 6c and f). We observe thermal conductance enhancement by introducing defect scattering across different temperatures and sizes.

Fig. 6. Thermal conductance of 4H-SiC and wurtzite GaN systems. **a** Thermal conductance of 4H-SiC systems at different temperatures. **b** Thermal conductance of 4H-SiC systems at 100 K with a fixed length of the heating zone (10 nm) and different lengths of the substrate. **c** Thermal conductance of 4H-SiC systems at 100 K with a

fixed length of the substrate and different lengths of the heating zone. **d** Thermal conductance of wurtzite GaN systems at different temperatures. **e** Thermal conductance of wurtzite GaN systems at 100 K with a fixed length of the heating zone (10 nm) and different lengths of the substrate. **f** Thermal conductance of wurtzite GaN systems at 100 K with a fixed length of the substrate and different lengths of the heating zone.”

“Our analysis so far has assumed that mode-level heat generation is proportional to the modal heat capacity (spectral equilibrium). However, we also investigate the thermal conductance with considering the over-population of optical phonons (spectral nonequilibrium). We conduct a rigorous electron-phonon coupling calculation for Si to determine the mode-level heat generation (see Supplementary Note S7). As shown in Fig. 7a, compared with cases for heat generation with spectral equilibrium, we actually find more significant thermal conductance enhancement by introducing defect scattering for heat generation with spectral nonequilibrium. At 100 K, the maximum thermal conductance enhancement reaches 338%. The underlying mechanism is that defect scattering not only manipulates the directional nonequilibrium but also the spectral nonequilibrium (see Supplementary Note S7).

Fig. 7. Thermal conductance for heat generation with spectral nonequilibrium and diffusely reflecting boundary (Si systems). **a** Thermal conductance for heat generation with spectral nonequilibrium. **b** Thermal conductance when adopting the diffusely reflecting boundary. Si systems with 10 nm heating zone and 40 nm substrate are studied.”

(iii) The results with regard to the over-population of optical phonons are interesting, as the inclusion of defect scattering leads to a more significant increase in thermal conductance. This is based on an assumption that optical phonons obtain all energy. Recent electron-phonon coupling calculations show that some acoustic phonons can still obtain energy from electrons. It would be better if the authors discuss how significant this effect can be if the optical phonons obtain less energy (for example, according to rigorous electron-phonon coupling calculation).

Response: Thank for the reviewer's insightful suggestion. We have conducted thorough electron-phonon coupling calculations for silicon (Si) to obtain a more accurate mode-level heat generation. The rigorous electron-phonon coupling calculations are implemented in the electron-phonon Wannier (EPW) package [1]. The electron-phonon coupling matrix elements are firstly calculated on the coarse meshes and are then interpolated to $100 \times 100 \times 100$ k-point and $60 \times 60 \times 60$ q-point meshes to calculate the electron-phonon energy generation rate with our modified codes. The calculated mode-level heat generation is shown in Fig. R1. There are several peaks of the heat generation, which means that electrons tend to transfer energy to specific phonon modes, especially for some optical phonon modes. Meanwhile, some acoustic phonon modes also have received energy from electrons.

As shown in Fig. R2, when compared to the scenario where optical phonons receive all the energy, our rigorous electron-phonon coupling analysis indicates a slightly reduced over-population of optical phonons, resulting in a minor decrease in thermal conductance enhancement through defect scattering. However, with this spectral nonequilibrium, the inclusion of defect scattering can still lead to a significantly more pronounced increase in thermal conductance compared to scenarios where heat generation follows spectral equilibrium.

Fig. R1 Phonon dispersion and the mode-level heat generation of Si from rigorous electron-phonon coupling calculation.

Fig. R2 Thermal conductance for different heat generation (Si systems with 10 nm heating zone and 40 nm substrate) at 300 K. The red dotted line represents calculated results when adopting spectral equilibrium heat generation. The light blue dashed line represents calculated results when assuming only phonons with frequencies larger than 400 cm^{-1} (optical phonons) receive energy from heat generation. The dark blue solid line represents calculated results when adopting heat generation from rigorous electron-phonon coupling calculations.

We also recognize the importance of rigorously calculating electron-phonon coupling to achieve a more accurate heat generation among different phonon modes. Consequently, we have incorporated these results into the updated manuscript (Page 16:

“Our analysis so far has assumed that mode-level heat generation is proportional to the modal heat capacity (spectral equilibrium). However, we also investigate the thermal conductance with considering the over-population of optical phonons (spectral nonequilibrium). We conduct a rigorous electron-phonon coupling calculation for Si to determine the mode-level heat generation (see Supplementary Note S7). As shown in Fig. 7a, compared with cases for heat generation with spectral equilibrium, we actually find more significant thermal conductance enhancement by introducing defect scattering for heat generation with spectral nonequilibrium. At 100 K, the maximum thermal conductance enhancement reaches 338%. The underlying mechanism is that defect scattering not only manipulates the directional nonequilibrium but also the spectral nonequilibrium (see Supplementary Note S7).”

In the updated Supplementary Materials:

“Recent studies reveal that in transistors or Raman measurements, the heat generation due to electron-phonon interactions is in spectral nonequilibrium, i.e., optical phonons (phonons with high frequency) tend to receive much more energy than acoustic phonons (phonons with low frequency) [8-11]. To study how this spectral nonequilibrium affects thermal conductance enhancement by defect scattering, we perform mode-level phonon BTE calculations with first-principles phonon properties for Si systems with a 10 nm heating zone and a 40 nm substrate. We have also conducted rigorous electron-phonon coupling calculations using the electron-phonon Wannier (EPW) package [12]. The electron-phonon coupling matrix elements are firstly calculated on the coarse meshes and are then interpolated to $100 \times 100 \times 100$ k-point and $60 \times 60 \times 60$ q-point meshes to calculate the electron-phonon energy generation rate with our modified codes. The calculated mode-level heat generation is shown in Fig. S10a. There are several peaks of the heat generation, which means that electrons tend to transfer energy to specific phonon modes, especially for some optical phonon modes. Meanwhile, some acoustic phonon modes also have received energy from electrons.

To quantify phonon spectral nonequilibrium, the spectral phonon temperature is usually adopted [9-11,13]. The definition of the spectral phonon temperature is

. When the phonons with different frequencies are in equilibrium, they have the same spectral phonon temperature. In the ballistic regime, since optical phonons receive all the energy and have poor thermal transport efficiency [9,14], optical phonons have very high temperatures that are much higher than acoustic phonons (as shown in Fig. S10b). When defect scattering is induced, the spectral nonequilibrium among phonons is largely reduced (as shown in Fig. S10b). Since acoustic phonons have higher thermal transport efficiency [9,14], the temperature is lower when the spectral nonequilibrium is smaller with a fixed heat flux.

Fig. S10 | a Phonon dispersion and the mode-level heat generation of Si from rigorous electron-phonon coupling calculation. **b** Spectral phonon temperature distribution at the interface between the heating zone and the substrate at 100 K (Si systems with 10 nm heating zone and 40 nm substrate). ”

(iv) There is a mistake in Fig. 4c where the center of the red arrow does not align with the gray line, which should be corrected.

Response: Thanks for the comment. We have corrected this mistake in Fig. 4c.

(v) In Wurtzite GaN and 4H-SiC systems, inducing defect scattering seems to monotonously increase the thermal conductance. If one further increases the defect scattering in Wurtzite GaN and 4H-SiC systems, will the thermal conductance decrease like that in the silicon system? It is better if some discussions are provided.

Response: Thanks for the comment. According to the reviewer’s suggestion, we have

modeled wurtzite GaN and 4H-SiC systems with more defect scattering. Specifically, as the common dopant for GaN, ^{24}Mg atoms are introduced as defects. Similarly, ^{10}B atoms are introduced in 4H-SiC. As shown in Fig. R3 and Fig. R4, it can be seen that the thermal conductance has decreased at high defect concentrations, similar to that in the silicon system.

Fig. R3 Thermal conductance at different temperatures when doping ^{24}Mg atoms in the GaN.

Fig. R4 Thermal conductance at different temperatures when doping ^{10}B atoms in the 4H-SiC.

We also added the discussions about the wurtzite GaN and 4H-SiC systems with more defect scattering in the updated manuscript (Page 15):

“Additionally, we explore the impact of ^{24}Mg impurities in GaN and ^{10}B impurities in SiC, as detailed in Supplementary Notes S5 and S6. Phonon properties are obtained from first-principles calculations (see Supplementary Notes S5 and S6). Our

investigation encompasses varying temperatures (Fig. 6a and d), substrate lengths (Fig. 6b and e), and heating zone lengths (Fig. 6c and f). We observe thermal conductance enhancement by introducing defect scattering across different temperatures and sizes.”

In the updated Supplementary Materials:

“Wurtzite GaN is widely used in power electronics [6]. We investigate thermal conductance enhancement in GaN systems with defects. We first extract phonon properties from first-principles calculations with the quantum phonon population (Bose-Einstein distribution). A supercell of $4 \times 4 \times 4$ and the fifth nearest atom neighbor is considered to obtain the third-order anharmonic interatomic force constants. The thermal conductivity of GaN is calculated based on the single-mode relaxation time approximation method. We use $40 \times 40 \times 40$ q-points for all temperatures to sample the Brillouin zone. The bulk thermal conductivity of GaN for different temperatures is shown in Fig. S8a. Our results agree well with those in the literature [2]. To estimate the scattering from defects, we adopt the Tamura model [7]. We adopt the mode-level phonon BTE calculations to investigate the thermal conductance enhancement by doping ^{24}Mg atoms to replace Ga atoms (system in Fig. 1(a)) with different temperatures as shown in Fig. S8b. In the main text, we introduced doping with ^{71}Ga isotopes and observed a continuous and monotonic increase in thermal conductance. This behavior can be attributed to the small mass difference between ^{71}Ga and ^{69}Ga isotopes. In contrast, a significant mass difference exists between ^{24}Mg and Ga atoms, leading to more pronounced defect scattering induced by ^{24}Mg compared to ^{71}Ga isotopes. Consequently, the thermal conductance demonstrates an initial increase followed by a subsequent decrease, as illustrated in Figure S8b.

Fig. S8 | Results of wurtzite GaN systems. **a** Bulk thermal conductivity of GaN. The symbols correspond to values from the reference [2]. **b** Thermal conductance at different temperatures when doping ^{24}Mg in GaN.

4H-SiC is also widely used in power electronics [6]. We investigate the thermal conductance enhancement in SiC systems with C^{14} isotopes. We first extract phonon properties from first-principles calculations with the quantum phonon population (Bose-Einstein distribution). A supercell of $4 \times 4 \times 2$ and the fourth nearest atom neighbor is considered to obtain the third-order anharmonic interatomic force constants. The thermal conductivity of SiC is calculated based on the single-mode relaxation time approximation method. We use $23 \times 23 \times 7$ q-points for all temperatures to sample the Brillouin zone, which is consistent with the reference [3]. The bulk thermal conductivity of SiC for different temperatures is shown in Fig. S9a. Our results agree well with those in the literature [3]. To estimate the scattering from defects, we adopt the Tamura model [7]. We adopt the mode-level phonon BTE calculations to investigate the thermal conductance enhancement by doping ^{10}B atoms to replace Si atoms (system in Fig. 1(a)) with different temperatures as shown in Fig. S9b. In the main text, we introduced doping with ^{14}C isotopes and observed a continuous and monotonic increase in thermal conductance. This behavior can be attributed to the small mass difference between ^{14}C and ^{12}C isotopes. In contrast, a significant mass difference exists between ^{10}B and Si atoms, leading to more pronounced defect scattering. Consequently, the thermal conductance demonstrates an initial increase followed by a subsequent decrease, as illustrated in Figure S9b.

Fig. S9 | Results for 4H-SiC systems. **a** Bulk thermal conductivity of SiC. The symbols correspond to values from the reference [3]. **b** Thermal conductance at different temperatures when doping ^{10}B in SiC.”

Response to Reviewer #2:

This manuscript reports on a study of the effects of defect scattering in the heating zone on the calculated thermal conductance using molecular dynamics simulation and numerically solving the Boltzmann transport equation. Contrary to the common expectation of enhanced resistance from defects, the results here show that defects in the heat zone could increase the derived thermal conductance. The authors attributed the enhanced thermal conductance to the defect scattering induced spatial redistribution of phonon energy and introduced the concept of directional non-equilibrium.

The authors have conducted extensive modeling to prove their arguments and the referee feels that the disclosed underlying physics is interesting. However, the referee is not convinced that the observed phenomena are not due to numerical artifacts in the simulations. Therefore, the manuscript has to be reconsidered after the authors address the following concerns.

Response: We sincerely appreciate the reviewer's thoughtful comments of our manuscript and extend our gratitude for the valuable feedback provided. The reviewer's interest in the extensive modeling supporting our findings and positive assessment of the underlying physics is greatly acknowledged. In response to the specific concerns regarding the potential influence of numerical artifacts on the observed phenomena, we have carefully addressed the issues raised and implemented detailed modifications to the manuscript. Following thorough examination and multiple verifications across different cases, we affirm that the observed phenomena are not a result of numerical artifacts. We believe that these revisions and additions not only address the concerns but also enhance the clarity of our discussion, thereby fortifying the overall reliability of our research. It is important to emphasize that our computations adhere to rigorous and widely accepted standards, and are grounded in realistic physics. Furthermore, we would like to highlight that the observed novel phenomenon shows promise for experimental validation, particularly at low temperatures. The reviewer's insights have been invaluable, and we hope that our efforts in addressing these concerns are satisfactory.

(i) The authors claimed that “since both MD and BTE can show that introducing defect scattering can lead to enhanced thermal transport, it is clearly not caused by simulation artifacts.” The referee disagrees with this argument because in the BTE calculation, the boundary conditions are set to match the transport mechanism in the MD. As such, it is not a surprise that the MD and BTE yielded similar trends.

Response: Thanks for the valuable feedback. We appreciate the meticulous attention on our work and would like to address the concerns regarding potential simulation artifacts impacting our reported results.

Regarding the agreement between the two methods, it is essential to note that we did not artificially match the boundaries between BTE and MD. Instead, this agreement naturally arises from the consistent thermal transport mechanisms underlying both methods. MD operates in the real space, whereas BTE employs the phonon space to define boundaries. Our simulations utilized standard settings in BTE, including adiabatic boundary and thermalizing boundary, as well as in MD including fixed atoms, and commonly used MD thermostats such as Langevin and Nose-Hoover chain thermostats. No special modifications were introduced in either method.

It should be noted that achieving quantitative agreement between MD and BTE is a difficult task that extends beyond boundary conditions. It necessitates a comprehensive understanding of thermal reservoirs, boundary conditions, accurate phonon property calculations, and precise computation processes for both methods. Since MD works with atoms while BTE works with phonons, the alignment of MD and BTE has been a long-standing issue in nanoscale thermal simulation [2,3]. Our recent research has also highlighted these challenges [4]. In this study, we have successfully achieved quantitative alignment between the two methods (as illustrated in Fig. 5 in the manuscript), providing robust evidence for the accuracy of our calculations and supporting the credibility of our findings.

Additionally, we explored alternative boundary conditions, such as eliminating fixed atoms in MD and adopting periodic boundaries in a symmetrical system, as shown in Fig. R5b. Interestingly, even without fixed atoms, we observed an increase in thermal conductance due to defect scattering (Fig. R6). This phenomenon still exists when

studying rough surfaces (as addressed in concern iii). The observed phenomena persisted under these different systems, giving us more confidence on the validity of our findings instead of numerical artifacts.

Fig. R5 **a** Simulation system studied in MD in the manuscript. **b** Symmetrical simulation system with periodic boundaries instead of adopting fixed atoms in MD. The substrate is pure Si and the heating zone is Si with Ge impurities occupying random sites.

Fig. R6 Thermal conductance results from MD for simulation systems shown in Fig. R5.

To enhance clarity and avoid potential misunderstandings, we have also revised the respective sentences in the updated manuscript (Page 3 and Page 8):

“We also try a system without fixed atoms by using periodic boundaries and find the negligible influence of fixed atoms (see Supplementary Note S1).”

“Both MD and BTE show that introducing defect scattering can lead to enhanced thermal transport. As will be shown later, with consideration of the mode-level phonon properties, the two methods quantitatively match, providing robust evidence for the accuracy of our calculations and supporting the validity of our findings.”

In the updated Supplementary Materials:

“We also eliminate fixed atoms in MD simulations and adopt periodic boundaries, as shown in Fig. S2. Without fixed atoms, there is also an increase in thermal conductance due to defect scattering (Fig. S3), indicating that fixed atoms have a negligible impact on the results.

Fig. S2 | **a** Simulation system studied in MD in the manuscript. **b** Symmetrical simulation system with periodic boundaries instead of adopting fixed atoms in MD. The substrate is pure Si and the heating zone is Si with Ge impurities occupying random sites.

Fig. S3 | Thermal conductance results from MD for simulation systems shown in Fig. S2.”

(ii) The referee has the concern that the effect is actually induced by simulation artifacts, i.e., the way that heat is added to the heating zone. In fact, the authors stated “When the scattering is rare (i.e., in the ballistic regime) inside the uniform volumetric heating zone (Fig. 4d), the phonon mode propagating oblique to the z direction (mode 2) travels a much longer distance in the heating zone than the mode propagating along the z direction (mode 1), and thus receives a much larger amount of energy.” The referee feels that this explanation exactly suggests that the effect is indeed due to how energy is added to the heating zone.

Response: Thanks for the comments. We appreciate the attention to detail and would like to address the concerns regarding the potential influence of simulation artifacts on our reported results. It is important to clarify that the explanation presented in our manuscript is grounded in our understanding of the underlying phenomena and is not contingent on the simulation settings. In our simulations, we employed a standard thermal reservoir for the heat source in MD known as the Nose-Hoover thermostat. This choice corresponds to a scenario of uniform volumetric heating and aligns with established practices in the field, as verified by both Boltzmann Transport Equation (BTE) and experimental validations [5,6]. It is essential to remark that artificially introducing directional phonon heating is technically impossible in MD simulations due

to its atomic vibration nature. Instead, MD only permits control over the average temperature in the heating zone by scaling the atomic velocities. In the BTE calculations, we utilized a widely accepted volumetric heating method [7]. The resulting directional heating observed from both methods is an outcome deeply rooted in the principles of ballistic transport, and not a prescribed condition. We explain it by the difference in traveling distances of different phonon modes. As such, we believe that the observed effect is a legitimate physical phenomenon under consideration, free from numerical artifacts.

(iii) The authors set the reflection at the upper boundary as specular, which helps maintain the direction of phonons from the normal direction of the boundary. Based on the referee's understanding of the manuscript, if the upper boundary is set as fully diffuse reflection/scattering, the effect will be significantly diminished, which would mean that indeed the phenomenon is a simulation artifact. Similarly, in MD, if the interface between the fixed atoms and the heating zone is set to be rough, the effects should be less significant.

Response: Thanks for the insightful comments and suggestions regarding the simulation setup in our manuscript. In response to the recommendation, we conducted additional simulations by modifying the upper boundary condition from specular to diffuse reflection in BTE as shown in Fig. R7b. The results show a slight reduction in thermal conductance enhancement compared to cases with specular boundaries (Fig. R7c). The reduction is consistent with the reviewer's point. However, a distinct thermal conductance increase through introduced defects is still observed. This indicates that diffuse boundary scattering does not remove directional nonequilibrium to a great degree in the defect-free case, and the effect still well persists.

It should also be noted that both specular and diffuse reflection boundaries could be realized in experiments [8,9]. Consequently, our research provides guidance on maximizing the enhancement of thermal conductance through defect scattering in experiments, favoring the use of specular boundaries.

Fig. R7 **a** Simulation system studied in BTE with a specular reflection boundary in the manuscript. **b** Simulation system with a diffuse reflection boundary. **c** Thermal conductance for specular and diffuse reflection boundary shown in **a** and **b** (Si systems with 10 nm heating zone and 40 nm substrate) from phonon BTE calculations.

We have also added a few sentences to specify this point in the updated manuscript (Page 17):

“In all the aforementioned discussions, we consistently applied a specular reflection boundary condition to the upper boundary across all systems. To show the robustness of our findings, we also explored systems with diffuse reflection boundary conditions (see Methods). As illustrated in Fig. 7b, a decrease in thermal conductance enhancement is observed when compared to cases with specular boundaries. This is expected as diffuse boundary conditions are similar to defect scattering and tend to randomize the phonon directions, thereby contributing to the reduction of directional nonequilibrium. It is noteworthy that both specular and diffuse reflection boundaries can be experimentally implemented [42,43], and the boundary in reality is most likely partially specular and partially diffuse. Consequently, our research offers valuable insights into optimizing thermal conductance enhancement through impurity scattering in experimental settings, with a preference for the utilization of more specular boundaries.

Fig. 7. Thermal conductance for heat generation with spectral nonequilibrium and diffusely reflecting boundary (Si systems). **a** Thermal conductance for heat generation with spectral nonequilibrium. **b** Thermal conductance when adopting the diffusely reflecting boundary. Si systems with 10 nm heating zone and 40 nm substrate are studied.”

In the updated Methods (Page 21):

“The diffusely reflecting boundary condition is another type of adiabatic boundary condition, in which the energy of phonon reflected from the boundary is the same along each direction, i.e.,

where \hat{n} is the exterior normal unit vector of the boundary.”

(iv) The definition of $\Delta T_{\text{heat}}^{\text{bar}}$ in the first line of Page 9 is confusing.

Response: Thanks for the reviewer for bringing attention to the definition of $\Delta T_{\text{heat}}^{\text{bar}}$ in the first line of Page 9. We appreciate the reviewer’s valuable feedback and have clarified its purpose and meaning. In our revised manuscript, we explicitly define $\Delta T_{\text{heat}}^{\text{bar}}$ as the average temperature rise within the heat source relative to the substrate, with the calculation expressed as $\Delta T_{\text{heat}}^{\text{bar}} = \frac{\int_{V_{\text{HS}}} T dV}{V_{\text{HS}}}$. This definition aligns with the average temperature rise within the heat source as defined in MD simulations. We have modified the corresponding part in the updated manuscript (Page 9):

“ , where denotes the average temperature drop inside the heating zone relative to the heating zone-substrate interface, with the calculation expressed as ”

(v) In the derivation of thermal conductance from MD, the authors adopted an equation that includes the effects of the heat source and sink. Therefore, the derived the conductance includes the boundary effect. What if the authors adopt the conductance as the ratio of heat flux over the temperature gradient in the substrate?

Response: We appreciate the insightful comment regarding the definition of thermal conductance. In response to the suggestion about adopting the conductance as the ratio of heat flux over the temperature gradient in the substrate, we find this proposal intriguing. At the ballistic transport regime, temperature is not the macroscopic thermodynamic definition and primarily serves as a representation of total internal energy density. Previous studies have widely observed nonlinear temperature distributions even under constant heat flow [10]. Particularly in the ballistic limit, there can be heat flow even when there is no temperature gradient. In such systems, the temperature gradient cannot be well-defined. Our recent research also supports this point [4].

The thermal conductance defined in this study is an effective thermal conductance used to characterize systems where better heat dissipation leads to lower temperature rise under the same heating conditions. It aligns with the definition of device thermal resistance used in the field of transistor heat dissipation [11]. Moreover, in the ballistic limit, this definition can be unified with the Landauer formula [12]. Regarding the reviewer's remark about the inclusion of heat source and sink effects, as shown in Fig. 2 in the manuscript, inducing defects not only results in a temperature decrease within the heating zone but also in the substrate. Meanwhile, it is noteworthy that the inclusion of heat sources constitutes a key innovation in our work. In past studies for MD and BTE, the heat source and sink have often been ignored [13,14]. Our research emphasizes the importance of the heat source, showing its significant impact on heat

transfer.

To enhance clarity and avoid potential misunderstandings, we also add sentences in the updated manuscript (Page 5):

“To quantify the performance of the heat dissipation, we define the thermal conductance K_{th} for this system, where Q_z is the heat flux along the z direction in the substrate and T_{avg} is the average temperature in the heating zone [16]. The definition of this thermal conductance is similar to the widely used thermal resistance in the heat dissipation of transistors, representing that, under the same heat generation, the lower the device temperature rise, the smaller the thermal resistance, and the greater the thermal conductance [17-19].”

(vi) Since there is no scattering in the substrate in the BTE modeling, the definition of ΔT_{sub} is confusing. It is a temperature jump at the lower boundary instead of a temperature drop across the substrate.

Response: Thanks for the valuable feedback on our manuscript. In response to the suggestion, we have made appropriate revisions in the updated manuscript (Page 9) to distinguish between the temperature drop across the substrate and the temperature jump at the lower boundary:

“For these three cases, there is a temperature drop inside the heating zone and another temperature drop from the heating zone-substrate interface to the bottom boundary. Note that the temperature inside the substrate is constant since the scattering inside the substrate is neglected [30]. The temperature drop from the heating zone-substrate interface to the bottom boundary is equal to the temperature jump at the boundary in present cases. We divide the total temperature drop into two parts: the temperature drop inside the heating zone ΔT_{hz} and the temperature drop from the heating zone-substrate interface to the bottom boundary ΔT_{sub} , i.e. temperature jump at the boundary (Fig. 4a).”

(vii) Under special conditions, defects can actually lead to enhanced thermal conductance as demonstrated by quite a few experiments. The authors should discuss their results with respect to those experiment results and provide perspectives on the different transport mechanisms.

Response: Thanks for the reviewer's insightful comment. We have noticed that Zhang et al. [15] recently utilized defects to assist phonon transport through kinked nanowires. Another work published during the peer review process of this study by Liu et al. [16] involves doping oxygen atoms in van der Waals crystal TiS_3 nanoribbons to induce lattice contraction and increase Young's modulus, leading to enhanced thermal conductivity. These studies differ in fundamentals from the current study that focused on reducing directional nonequilibrium through defect scattering. Nevertheless, the potential coexistence of these effects provides further viable means for manipulating heat transfer at the micro- and nanoscale.

Furthermore, we believe the observed phenomenon holds promise for experimental validation. Our system has real-world counterparts, where the volumetric heating method is aligned with optical and electrical heating in experiments [17,18]. The heat sink corresponds to the thermal reservoir in the heat bridge method [19,20]. Our discussions on the influence factors also provide guidance on maximizing the enhancement of thermal conductance through defect scattering in experiments. Currently, spectral phonon nonequilibrium has been experimentally observed, and directional nonequilibrium systems may pose more stringent requirements, necessitating precise control of the heat source. We are actively working on achieving experimental enhancement of thermal conductance through defect scattering under spectral nonequilibrium.

We also add some discussions in the updated manuscript (Pages 12 and 17-18):

“Note that Zhang et al. [33] recently utilized defects to redirect phonons and assist phonon transport through kinked nanowires, which differs in principle from this study that focused on reducing directional nonequilibrium through defect scattering.”

“We have also noticed another work by Liu et al. [45] published during the peer review process of this study, which doped oxygen atoms in van der Waals crystal TiS_3

nanoribbons to induce lattice contraction and increase Young's modulus, leading to enhanced thermal conductivity. Distinct from inducing defects to migrate directional nonequilibrium in this study that reveals a general mechanism to enhance thermal transport, the strategy provided by Liu et al. is available for limited material systems. Nevertheless, the potential coexistence of these effects provides further viable approaches for manipulating heat transfer at the micro- and nanoscale. The observed phenomenon in this study also holds promise for experimental validation. Our system has real-world counterparts, where the volumetric heating method aligns with optical and electrical heating in experiments [35,40]. The corresponding heat sink corresponds to the thermal reservoir in the heat bridge method [46,47]. Our discussions on the influencing factors also provide guidance on maximizing the enhancement of thermal conductance through defect scattering in experiments (at lower temperatures, with smaller sizes, and with a specular upper surface.)”

Response to Reviewer #3:

In this paper, the authors show that the defect-free heating zone overpopulates oblique-propagating phonons, while introducing defects would redirect phonons randomly to restore directional equilibrium. They demonstrate that defect scattering can enable such thermal transport enhancement in a wide range of temperatures, and offer an unconventional strategy for enhancing thermal transport. The results are interesting and important. There are some issues for authors to address:

Response: We are grateful to the reviewer for the positive evaluation.

(i) A cross-sectional area of 8×8 is not large, whether there is an effect of phonon-boundary scattering on thermal conductivity needs to be further demonstrated.

Response: Thank for the reviewer's insightful comment regarding the cross-sectional area and the potential effect of phonon-boundary scattering. Since periodic boundary conditions are applied in lateral directions as shown in Fig. 1 in the manuscript, we actually model a one-dimensional system. To check whether a cross-sectional area of 8×8 is sufficient, we further increase the cross-sectional area (up to 24×24), as shown in Fig. R8. It can be seen that the results do not change, suggesting that phonon-boundary scattering does not play a significant role in the current system.

Fig. R8 Thermal conductance from MD simulations for a cross-sectional area of 8×8 , 16×16 , and 24×24 -unit cells.

We have also added a few sentences in the updated manuscript (Page 3):

“After a convergence test (see Supplementary Note S1), a cross-sectional area of 8×8 -unit cells is considered and periodic boundary conditions are applied in lateral directions.”

We also added a few sentences in the updated Supplementary Materials:

“To check whether a cross-sectional area of 8×8 -unit cells is sufficient, we further increase the cross-sectional area (up to 24×24 -unit cells), as shown in Fig. S1. It can be seen that the results do not change, suggesting that phonon-boundary scattering does not play a significant role in the current system.

Fig. S1 | Thermal conductance from MD simulations for a cross-sectional area of 8×8 , 16×16 , and 24×24 -unit cells.”

(ii) How do phonons with different vibrational frequencies contribute differently to the thermal conductivity? The PDOS of phonons with different doping concentrations are useful to complement the phonon transport mechanism.

Response: Thank for the reviewer’s insightful comment regarding the DOS of phonons with different defect concentrations for complementing the phonon transport mechanism. The extracted DOS from MD simulations for Si under different Ge concentrations is shown in Fig. R9 below. It shows that the defect concentration has a minor impact on the DOS, most likely attributed to the low defect concentration. How

different phonons contribute to the thermal transport can be analyzed from the phonon scattering rate or phonon mean free path with different frequencies under different defect concentrations as shown in Fig. R10. It can be seen that low-frequency acoustic phonons have a relatively large contribution to the thermal conductivity. A small defect concentration leads to significant decreases in the mean free path of the phonon spectrum, especially at low temperatures. The decreased phonon mean free paths in the heating zone weaken the directional phonon nonequilibrium and therefore lead to enhanced thermal conductance.

Fig. R9 The phonon density of states for Si under different doped Ge concentrations.

Fig. R10 Phonon mean free path distribution of Si with Ge impurities calculated from first-principles calculations at **a**, 50 K, **b**, 100 K, **c**, 300 K, and **d**, 400 K.

We have also added some discussions in the updated Supplementary Materials:

“Figure S6 shows the phonon density of state (DOS) of Si with Ge impurities. It shows that the concentration has a minor impact on the DOS, most likely attributed to the low defect concentration. How different phonons contribute to the thermal transport can be analyzed from the phonon scattering rate or the phonon mean free path with different frequencies from first-principles calculations as shown in Fig. S7.

Fig. S6 The phonon density of states for Si under different doped Ge concentrations.

Fig. S7 | Phonon mean free path distribution of Si with Ge impurities calculated from first-principles calculations at a, 50 K, b, 100 K, c, 300 K, and d, 400 K.

(iii) In Fig. 5c, why does the enhancement of G first increase and then decrease as the substrate length increases?

Response: Thank for the reviewer's insightful comment. As shown in Fig. 4a and b in the manuscript, introducing defects in the heating zone can decrease the ΔT_{sub} (i.e., the temperature drop from the heating zone-substrate interface to the bottom boundary), and increase the ΔT_{heat} (i.e., the temperature drop inside the heating zone). This is also illustrated in Fig. R11, where Fig. R11a and b show the ΔT_{sub} and ΔT_{heat} at different substrate lengths L_{sub} for pure silicon system and the system with 0.5% Ge in the heating zone, respectively ($L_{\text{heat}}=10$ nm and the total heat generation/heat flux is constant). Fig. R12 shows the ratio of decreased ΔT_{sub} and the ratio of increased ΔT_{heat} when comparing these two systems. Since the thermal conductance is given by $G = \frac{Q}{\Delta T}$, it is the significantly decreased ΔT_{sub} from introducing defects in the heating zone that leads to the overall thermal conductance enhancement as shown in Fig. 5c.

In response to the review's comment, there are two competing effects come into play when varying the length of the substrate L_{sub} . First, as shown by the red line in Fig. R12, when L_{sub} is large, the decrease of ΔT_{sub} will not be as pronounced as small L_{sub} cases, due to the fact that the doping defects in a relatively small heating zone become less important to the heat transfer in the entire system. Hence, the thermal conductance enhancement attenuates for large L_{sub} . Second, as shown by the blue line in Fig. R12, when L_{sub} is small, the increase of ΔT_{heat} becomes more significant. This may originate from the more important role of the boundary when L_{sub} is small, which can induce additional ballistic thermal resistance and lead to higher temperature rise [13]. As such, the thermal conductance enhancement also attenuates for small L_{sub} . Overall, these two competing effects determine that the most pronounced thermal conductance enhancement is obtained at a moderate substrate length.

Fig. R11 **a** The ΔT_{sub} at different substrate lengths L_{sub} for pure silicon system and the system with 0.5% Ge in the heating zone. **b** The ΔT_{heat} at different substrate lengths L_{sub} for pure silicon system and the system with 0.5% Ge in the heating zone.

Fig. R12 the ratio of decreased (increased) $\Delta T_{\text{sub}}(\Delta T_{\text{heat}})$ at different substrate lengths L_{sub} .

Reference

- [1] S. Poncé, E. R. Margine, C. Verdi, and F. Giustino, *Computer Physics Communications* **209**, 116 (2016).
- [2] J. E. Turney, E. S. Landry, A. J. H. McGaughey, and C. H. Amon, *Physical Review B* **79**, 064301 (2009).
- [3] X. Zhang, H. Xie, M. Hu, H. Bao, S. Yue, G. Qin, and G. Su, *Physical Review B* **89**, 054310 (2014).
- [4] Y. Hu, T. Feng, X. Gu, Z. Fan, X. Wang, M. Lundstrom, S. S. Shrestha, and H. Bao, *Physical Review B* **101**, 155308 (2020).
- [5] E. S. Landry and A. J. H. McGaughey, *Physical Review B* **80**, 165304 (2009).
- [6] D. P. Sellan, E. S. Landry, J. E. Turney, A. J. H. McGaughey, and C. H. Amon, *Physical Review B* **81**, 214305 (2010).
- [7] J. Kaiser, T. Feng, J. Maassen, X. Wang, X. Ruan, and M. Lundstrom, *Journal of Applied Physics* **121** (2017).
- [8] A. George *et al.*, *ACS Applied Materials & Interfaces* **11**, 12027 (2019).
- [9] T. J. Myers, J. A. Throckmorton, R. A. Borrelli, M. O'Sullivan, T. Hatwar, and S. M. George, *Applied Surface Science* **569**, 150878 (2021).
- [10] J. Xu, Y. Hu, X. Ruan, X. Wang, T. Feng, and H. Bao, *Physical Review B* **104**, 104310 (2021).
- [11] E. Pop, *Nano Research* **3**, 147 (2010).
- [12] Y. Imry and R. Landauer, *Reviews of Modern Physics* **71**, S306 (1999).
- [13] J. Maassen and M. Lundstrom, *Journal of Applied Physics* **117** (2015).
- [14] S. G. Volz and G. Chen, *Physical Review B* **61**, 2651 (2000).
- [15] Q. Zhang *et al.*, *Nano Letters* **17**, 3550 (2017).
- [16] C. Liu, C. Wu, X. Y. Tan, Y. Tao, Y. Zhang, D. Li, J. Yang, Q. Yan, and Y. Chen, *Nature Communications* **14**, 5597 (2023).
- [17] R. J. Warzoha *et al.*, *Journal of Electronic Packaging* **143** (2021).
- [18] H. Zobeiri, N. Hunter, R. Wang, T. Wang, and X. Wang, *Advanced Science* **8**, 2004712 (2021).
- [19] M. T. Pettes, I. Jo, Z. Yao, and L. Shi, *Nano Letters* **11**, 1195 (2011).
- [20] Z. Wang, R. Xie, C. T. Bui, D. Liu, X. Ni, B. Li, and J. T. L. Thong, *Nano Letters* **11**, 113 (2011).

REVIEWERS' COMMENTS

Reviewer #1 (Remarks to the Author):

I am very glad to see that in the updated revision, the authors have well clarified previous concerns and clearly answered all the comments, raised only from me but also from other referees. All the suggestions are also well incorporated to improve the revision.

Their observation of defect scattering enhanced nanoscale heat transport is significant novel, of broad interest and potential implications in industry. The evidence and explanation of the mechanism are also convincing. I recommend the immediate acceptance without hesitation.

Reviewer #2 (Remarks to the Author):

The authors have provided reasonable response to most concerns raised. This referee believes that the authors have done comprehensive simulations to support the interesting physical picture and recommends acceptance of the manuscript for publication. Hopefully, the new concept can be used to explain unexpected experimental observations, which will also provide experimental evidence to support the physical understanding disclosed in this manuscript.

Reviewer #3 (Remarks to the Author):

The authors revised the paper according to the review comments. I recommend the publication of the paper.

The authors sincerely appreciate the comments from all reviewers. The point-by-point responses to the comments are listed below.

Reviewer #1 (Remarks to the Author):

I am very glad to see that in the updated revision, the authors have well clarified previous concerns and clearly answered all the comments, raised only from me but also from other referees. All the suggestions are also well incorporated to improve the revision.

Their observation of defect scattering enhanced nanoscale heat transport is significant novel, of broad interest and potential implications in industry. The evidence and explanation of the mechanism are also convincing. I recommend the immediate acceptance without hesitation.

Response: We are grateful to the reviewer for the positive evaluation.

Reviewer #2 (Remarks to the Author):

The authors have provided reasonable response to most concerns raised. This referee believes that the authors have done comprehensive simulations to support the interesting physical picture and recommends acceptance of the manuscript for publication. Hopefully, the new concept can be used to explain unexpected experimental observations, which will also provide experimental evidence to support the physical understanding disclosed in this manuscript.

Response: We are grateful to the reviewer for the positive evaluation. The authors also look forward to further experimental validation of the findings in this study.

Reviewer #3 (Remarks to the Author):

The authors revised the paper according to the review comments. I recommend the publication of the paper.

Response: We are grateful to the reviewer for the positive evaluation.